

# Non-target screening of organic compounds in offshore produced water by GC×GC-MS

Sofie N. Bergfors, Khoa Huynh, Annette E. Jensen and Jonas Sundberg

Danish Hydrocarbon Research and Technology Centre, Technical University of Denmark, Kgs. Lyngby, Denmark

## ABSTRACT

Produced water is the largest by-product of oil and gas production. At off-shore installations, the produced water is typically reinjected or discharged into the sea. The water contains a complex mixture of dispersed and dissolved oil, solids and inorganic ions. A better understanding of its composition is fundamental to (1) improve environmental impact assessment tools and (2) develop more efficient water treatment technologies. The objective of the study was to screen produced water sampled from a producing field in the Danish region of the North Sea to identify any containing organic compounds. The samples were taken at a test separator and represent an unfiltered picture of the composition before cleaning procedures. The analytes were isolated by liquid-liquid extraction and derivatized using a silylation reagent to increase the volatility of oxygenated compounds. The final extracts were analyzed by comprehensive multi-dimensional gas chromatography coupled to a high-resolution mass spectrometer. A non-target processing workflow was implemented to extract features and quantify the confidence of library matches by correlation to retention indices and the presence of molecular ions. Approximately 120 unique compounds were identified across nine samples. Of those, 15 were present in all samples. The main types of compounds are aliphatic and aromatic carboxylic acids with a small fraction of hydrocarbons. The findings have implications for developing improved environmental impact assessment tools and water remediation technologies.

## INTRODUCTION

Produced water is by volume the largest byproduct in oil and gas production. For reservoirs on the Danish continental shelf, where production is often supported by water flooding, the volume of produced water typically exceeds the volume of oil (*The Danish Energy Agency, 2014*). This water contains a complex mixture of inorganic and organic compounds. Implementation of offshore water management strategies is challenging due to the large volumes and practical limitations on infrastructure. Current procedures consist of cleaning of the produced water, followed by discharge into the sea or reinjection in producing or disposal wells (*Lyngbaek & Blidegn, 1991*; *Røe & Johnsen, 1996*; *Røe Utvik, 1999*; *Durell et al., 2004*). For discharge, the OSPAR Convention, which

Corresponding author
Jonas Sundberg, jonsun@dtu.dk

was set in to force in 1998, sets a limit of 30 mg of dispersed oil per liter of water as an annual average (*OSPAR Commission, 2001*). There is currently no limit on the dissolved compounds. Furtthermore, detailed knowledge on their structure and abundance is lacking.

In simplified terms, produced water can be seen as the product of aqueous extraction of crude oil by mixing within the reservoir and during production. In reality, the process is complex and the composition is highly dependent on reservoir properties, field history and water injection strategies (*Bergfors, Schovsbo & Feilberg, 2020*). For example, studies have shown that salinity influences to the organic content, likely caused by a salting-out effect (*Barth, Borgund & Riis, 1990*; *Barth, 1991*; *Barth & Riis, 1992*; *Endo, Pfennigsdorff & Goss, 2012*; *Dudek et al., 2020*). At the producing platform, a gravitational separation of oil, water and gas is carried out. The resulting water phase contains dissolved and dispersed oil droplets. The latter is largely removed by physical methods, i.e. separators, hydrocyclones and gas flotation tanks/degassers implemented in series on the platform, typically resulting in levels below 10 mg/L at the discharge point (*Meldrum, 1988*; *Saththasivam, Loganathan & Sarp, 2016*; *Durdevic & Yang, 2018*). In contrast, the removal of dissolved organics requires chemical treatment, e.g. degradation via advanced oxidation processes (AOPs) (*Jiménez et al., 2019*; *Lin et al., 2020*; *Liu et al., 2021*). This is challenging to implement on offshore installations due to safety and the requirement of low residency times. Furthermore, the current generation of AOPs are mainly based on Fenton's reagent which produces toxic chlorinated byproducts when applied to saline water (*Kiwi, Lopez & Nadtochenko, 2000*; *De Laat & Le, 2006*; *Sirtori et al., 2012*). The goal of produced water management is not necessarily zero discharge but zero *harmful* discharge. Improved knowledge of its composition is thus not only beneficial to develop environmental impact assessment tools. The information may also be used to develop improved and targeted treatment methods, i.e specific removal of harmful components, and not total organic content. This has the potential to increase efficiency and reduce the cost of water management strategies.

Ultimately, the toxicity of the produced water is a complex function based on the structure and concentration of both the dispersed and dissolved oil components (*Strømgren et al., 1995*; *Bakke, Klungsøyr & Sanni, 2013*; *Niu et al., 2016*; *Lofthus et al., 2018*). Furthermore, inorganic species such as heavy metals should be taken into consideration. We argue that to fully understand the effect of organic compounds in produced water discharge, we must (1) carry out full structural characterization studies and (2) differentiate between dispersed and dissolved species and (3) develop methods to integrate the data into environmental impact assessment tools. Although numerous compounds in produced water have been identified as part of ongoing water quality studies, a large part of the chemical space remains unknown. Recent advances in analytical technology (i.e. multi-dimensional chromatography, high-resolution mass spectrometry) have allowed more in-depth studies of the organic composition (*Samanipour et al., 2019*, *2020*; *Sørensen et al., 2019*; *Dudek et al., 2020*). However, these studies mainly focus on general classes of naphthenic acids and are based on formula calculations of exact masses without further structural elucidations. *Sørensen et al. (2019)* described a

non-target screening study but did not provide details beyond compound classes (i.e. phenols vs. naphthalenes). Thus, a large knowledge gap in the identity of dispersed and dissolved organics still exists.

To narrow this gap we have carried out a non-target screening study. The objective was not to evaluate the efficiency of implemented cleaning procedures but to obtain an "unfiltered" picture of the water composition immediately after production. Several samples were obtained from the test separator of a producing platform in the Danish North Sea. The organics were extracted by liquid-liquid extraction (LLE) using dichloromethane (DCM). The extracts were derivatized to increase the volatility of oxygenated compounds and analyzed using comprehensive multidimensional gas chromatography (GC×GC) coupled with a high-resolution quadrupole-time of flight mass spectrometer (MS). One advantage of using GC-MS as compared to liquid-chromatography (LC)-MS is the possibility of identification by spectral matching using commercially available libraries. In contrast, LC-MS is significantly dependent on user-generated libraries based on reference compounds as the ionization efficiency is more dependent on experimental conditions (*Schymanski et al., 2015*). Approximately 1,500 compounds were detected per sample, with 120 unique compounds tentatively identified across all samples. To increase the confidence of library matches, a data processing workflow that implemented retention indices and accurate mass matching was implemented. The data were analyzed to obtain a broad knowledge of the types of compounds present, their relative abundance as well as common compounds present across all samples.

# MATERIALS & METHODS

## Chemicals and reagents

Benzoic acid, phenol, cycolhexanecarboxylic acid, octanoic acid, dichloromethane (LiChroSolv, Merck, Darmstadt, Germany), *n*-hexane (SupraSolv for gas chromatography MS, Merck, Darmstadt, Germany), magnesium sulfate (ReagentPlus, Redi-Dri, Sigma–Aldrich, St. Louis, MO, USA), N,O-bis(trimethylsilyl)trifluoroacetamide containing 1% of trimethylchlorosilane (BSTFA+TMCS, Supelco, Bellefonte, PA, USA) were used as received. Deuterated internal standards (naphthalene-$d_8$, acenaphthene-$d_{10}$, phenanthrene-$d_{10}$, chrysene-$d_{12}$, Supelco analytical standards) were used to monitor retention time shifts. $^1$D retention index calibration was performed using a linear C7 to C30 saturated alkanes mixture (Supelco, TraceCERT, Sigma–Aldrich, St. Louis, MO, USA). High purity water was obtained from a Milli-Q Advantage A10 unit. All chemicals and reagents were used as received.

## Sampling and sample preparation

Produced water samples were donated by Mærsk Oil & Gas (now Total E&P). The samples were acquired from a producing field (water-injected) located in the Danish region of the North Sea. The sampling campaign took place between June 2018 and February 2019. The samples were taken at irregular intervals with no replicates. Water samples were collected at the test separator on the production platform following protocols established

by the operator. The test separator is a simple gravity-based three-phase separator where the water settles below the oil and can be sampled. No cleaning or further processing of the samples was carried out at the platforms. The samples were received in plastic bottles (1 L), aliquoted (500 mL), and immediately treated with dilute hydrochloric acid (18%, one mL per 100 mL sample) for a final pH < 2. Nine samples from the received batch were selected for analysis. The samples were selected due to the absence of dispersed oil droplets as evaluated by visual inspection (using microscope). The samples were stored in the dark at 4 °C until extraction and analysis.

Three aliquots (50 mL) of each produced water sample were extracted using separate glassware. The aliquots were filtered through a 0.45 μm PTFE-filter to remove solids and particles. Each 50 mL aliquot was extracted with DCM (50 mL). The organic phase was washed with saturated aqueous sodium chloride (50 mL) and carefully removed in vacuo. The residue was reconstituted in $n$-hexane (1.5 mL) and dried over $MgSO_4$. An aliquot (1,000 μL) of the sample was transferred to a two mL vial, combined with deuterated internal standards (for monitoring of retention time stability), combined with BSTFA +TMCS (50 μL) and incubated at 70 °C for 30 min, whereafter, it was allowed to return to room temperature. The derivatized sample was further diluted 10-fold with $n$-hexane and immediately analyzed on the GC×GC-MS.

## Extraction recovery and reproducibility

A model produced water containing five representative model compounds (benzoic acid, phenol, 2-naphthoic acid, cyclohexanecarboxylic acid, and octanoic acid, each at 5 ppm, total organics 25 ppm) in synthetic formation water (see Supplemental Information) was prepared to establish variability in the sample preparation protocol and instrumental analysis. The concentration of total organics was chosen to emulate typical levels encountered at production platforms. The model water was extracted four times in two batches following an identical procedure as for the produced water samples. Three procedural blanks were prepared to establish background levels and experimental sources of contamination.

## GC×GC-MS analyses

GC×GC-MS data were acquired using an Agilent 7890B GC coupled to a 7200B QTOF high-resolution mass spectrometer (Agilent Technologies, Palo Alto, CA, USA). The system was equipped with a Zoex ZX-2 thermal modulator (Zoex Corporation, Houston, TX, USA). The separation was achieved using a combination of an Agilent DB-5MS UI ($^1$D, 30 m, 0.25 mm i.d., 0.25 μm $d_f$) and a Restek Rxi-17Sil MS ($^2$D, 2 m, 0.18 mm i.d., 0.18 μm $d_f$) capillary columns connected using a SilTite μ-union. The oven was temperature programmed as follows; 1 min hold-time at 50 °C, ramp to 320 °C, 3 °C $min^{-1}$ in constant flow mode (1 mL/min). The modulation period was set to 3 s with a 400 ms hot-jet duration. The MS transfer line was held at 280 °C. The MS acquired spectra in electron ionization mode (70 eV) with a mass range of 45–500 and an acquisition speed of 50 Hz. The instrument was operated in its 2 GHz sampling rate mode

to increase the dynamic range. Automatic mass calibration was performed for every 5$^{th}$ sample (approximately 7.5 h).

## Data processing

Baseline correction (*Reichenbach et al., 2003*), peak detection and library search were performed using GC Image v2.8.3 (Zoex, Houston, TX, USA). Mass spectra were matched against the NIST Mass Spectral Library (National Institute of Standards and Technology, 2017 edition; NIST v17, Software Version: 2.3) with a minimum match factor of 700. All compound tables were exported as comma-separated texts for external processing. A data processing workflow was implemented in Python (Python Software Foundation. Python Language Reference, version 3.7.4. Available at http://www.python.org). The script is available as Supplemental Information deposited in a Zenodo repository (DOI 10.5281/zenodo.4009045).

## RESULTS

### Sample preparation and analyses

Previous studies on dissolved organics in oilfield produced water have employed LLE or solid-phase extraction (SPE) (*Thomas et al., 2009*; *Kovalchik et al., 2017*; *Barros et al., 2018*; *Samanipour et al., 2019*). For produced water, LLE using DCM has been shown to have a similar recovery as SPE (*Samanipour et al., 2019*). The main difference was observed in the size distribution, where larger species tend to have a higher recovery using SPE due to their low solubility in DCM. LLE is non-discriminative in comparison to SPE which is used to fractionate compound classes based on adsorption characteristics. Thus, it allowed us to extract the broad range of organics present in produced water. Furthermore, the Norwegian Oil and Gas specialist network recommends LLE for the quantification of phenols in produced water (*Norwegian Oil & Gas, 2012*). Thus, it would allow us to 'see' what is missing using routine targeted analyses. Approximately 30 samples were received as part of the campaign.

The water samples had varying levels of oil, likely due to the sampling and status of the test separator. Samples that contained a clear separate layer of oil, including smaller amounts, were excluded from the study to minimize the risk of contamination (Fig. 1). Optical inspection of these samples showed that they contained large amounts of dispersed oil droplets, even when sampling below the oil layer (Fig. 2). After exclusion, nine samples were chosen to be included in the study with extraction and characterization. All samples were filtered through a 0.45 μm PTFE-filter to remove insoluble material and particles. A small aliquot (50 mL) of each sample was extracted in triplicate using an equivalent volume of DCM.

The dissolved components of produced water were largely expected to be oxygenated organics, i.e. alcohols (mainly phenols) and acids. To increase the volatility of the aforementioned compounds, the samples were silylated. After removal of the solvent in-vacuo, each sample was reconstituted in 1.5 mL of *n*-hexane. A one mL aliquot of the reconstituted extract was treated with BSTFA-TMCS and incubated at 70 °C for 30 min. After derivatization, the samples were further diluted 10-fold, meaning that the samples ultimately were concentrated approximately 3-fold (from 50 mL to 15 mL). The final
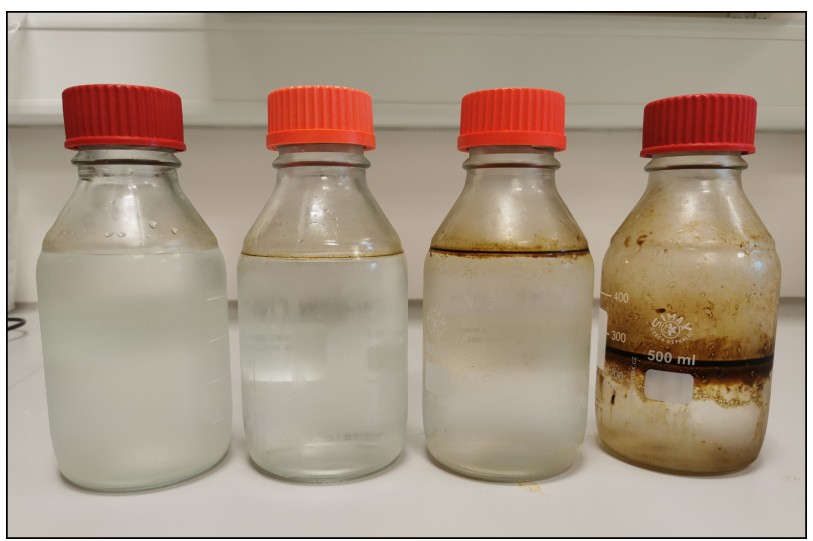

**Figure 1 Four representative samples showing the varying oil content.** Photo of four samples after decanting into glass bottles. Only visually clear samples were used in the study (two leftmost), while samples containing oil were excluded (two rightmost). The latter typically contained dispersed oil droplets in the aqueous phase as seen by optical microscopy (Fig. 2).

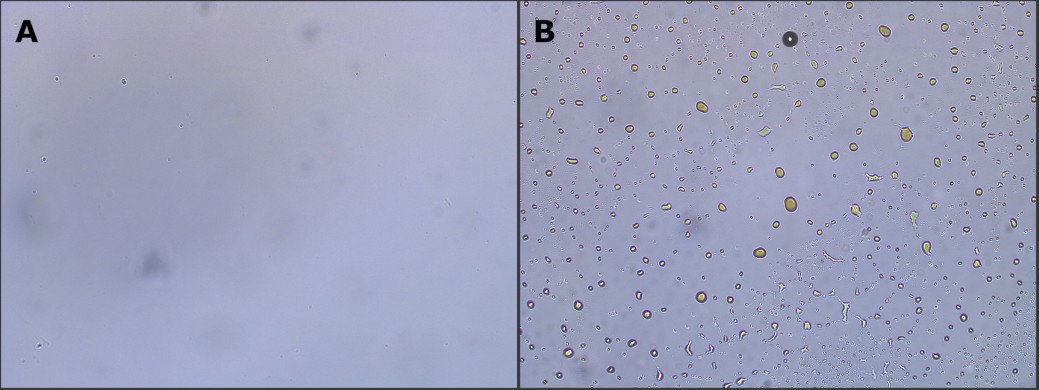

**Figure 2 Optical microscope image (10× magnification) of two different produced water samples.** The image shows the lack (A) or presence (B) of dispersed oil in the aqueous phase. Samples containing dispersed oil were excluded from the study.

concentration factor was chosen as a compromise between the detection of trace compounds and avoiding column and detector overload. Due to the large concentration range of our analytes, it would be beneficial to run each sample at multiple dilution factors. However, due to the long run time (90 min) and the number of extraction replicates (3) this was not feasible due to time constraints. It is important to remember that this strategy has two effects; (1) high concentration compounds may overload the detector with spectral skewing and poor library match as result, and (2) trace-compounds may be diluted to a level below the limit of detection.

Three procedural blanks were extracted and analyzed to determine background levels and potential sources of contamination. Minor amounts of fatty acids were detected in

**Table 1 Extraction recovery and retention time deviations for model and detected compounds.**

| Model compound | Recovery (%) | %RSD, model water | %RSD, standard | %RSD, real-life sample[*] |
|---|---|---|---|---|
| Phenol | 89.6 | 23.1 | 8.9 | 23.3 |
| Cyclohexanecarboxylic acid | 63.8 | 13.1 | 5.7 | 1.8 |
| Benzoic acid | 35.9 | 24.7 | 1.9 | 11.3 |
| Octanoic acid | 55.0 | 9.6 | 4.4 | 4.2 |
| 2-Naphthoic acid | 88.4 | 9.8 | 3.0 | 5.2 |
| **Internal standard** | **Avg. RT I (min.)** | **Avg. RT II (sec.)** | **RT I %RSD** | **RT II %RSD** |
| Naphthalene-D8 | 19.399 | 1.860 | 0.0004 | 0.8576 |
| Acenaphthene-D8 | 32.199 | 2.236 | 0.0002 | 0.6218 |
| Phenanthrene | 43.399 | 2.671 | 0.0002 | 0.5178 |

**Notes:**
[*] Peak volume values were extracted from a representative sample where the compound had been identified.
Recovery values are calculated as (peak volume$_{model\ water}$/peak volume$_{standard}$) × 100 ($n$ = 4 for model water and standard, 3 for real-life sample). Retention time deviations were calculated based on the retention of internal standards across all samples ($n$ = 27).

addition to a series of polysiloxanes (discrete peaks, not common column bleed). The source of the latter could not be identified but as it lacked retention in the $^2$D it did not interfere with our analytes of interest. To establish recovery and repeatability, a model produced water was prepared by spiking four organic acids and phenol into synthetic formation water (see Experimental section and Supplemental Data). All model compounds were detected in one or more produced water samples. The model water was extracted in four replicates. The recovery values were calculated based on peak volumes in comparison to those obtained by analyzing pure stock solutions (Table 1). The recovery varied from 36% for benzoic acid up to 90% for phenol. Considering the multi-step sample extraction, including a derivatization reaction, this was deemed acceptable. The relative standard deviation of the peak volumes (measured after baseline correction using the GC Image package) of model compounds as detected in a representative produced water sample varied from 2% to 23% (calculated from three extraction replicates).

## Chromatography

Crude oil is an ultracomplex mixture of saturated and aromatic hydrocarbons with a smaller fraction of N,S,O-containing compounds (*Marshall & Rodgers, 2004*; *Hsu et al., 2011*; *Palacio Lozano et al., 2020*). This complexity will be reduced but reflected in the produced water. Based on previous studies of Danish oils we know that the dominant oxygenated species belong to the O1 and O2 classes with a large diversity in aromaticity (*Sundberg & Feilberg, 2020*). Thus, the compositional variation in the produced water was assumed to be dominated by carbon number and level of saturation. Ultimately, the boiling point range was assumed to be larger than the variation in saturated versus aromatic structures. Therefore, we choose to use a non-polar column in the $^1$D (providing the highest separation power) with a shorter medium polarity column in the $^2$D. Conventional polar columns are based on polyethyleneglycol (PEG) chemistry and are incompatible with silylation reagents. Therefore, we choose to use a 50%-phenyl-type column where aromaticity is the largest factor affecting retention. By using this column combination, the

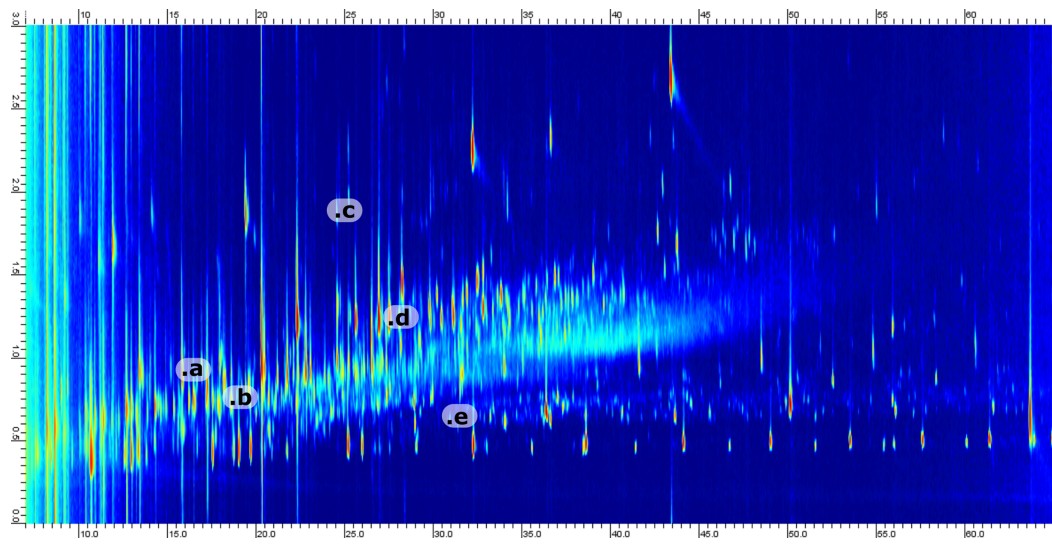

**Figure 3 Example 2D chromatogram showing the distribution of peaks with a heavy saturation of compounds between 0.5 and 1.5 s in the 2D.** Five representative compounds have been annotated in the chromatogram. The annotation corresponds to; (a) Cyclopentylcarboxylic acid, TMS derivative (b) Octanoic acid, TMS derivative (c) 1-Methylnaphthalene (d) Benzoic acid, 3-methyl-, trimethylsilyl ester (e) 2-Ethyl-1-decanol, TMS derivative.

retention in both dimensions will increase with aromaticity. For example, cyclohexane acetic acid has a retention of 17.2 min/0.84 s in $^1$D/$^2$D as compared to 24.6 min/1.33 s for benzeneacetic acid (measured as the corresponding trimethylsilyl esters). In contrast, alkylation of a core aromatic or saturated structure will only affect retention in the $^1$D. For example, phenol and its alkylated homologs (methyl, ethyl and propyl) have $^1$D retention times of 13.4, 16.9, 19.9 and 23.3 min, respectively, where the retention in $^2$D is within 0.89 to 0.95 s. By investigation of a typical chromatogram, two things become obvious; (1) the desired separation of saturates, mono- and diaromatics is achieved and (2) a large portion of the $^2$D space is relatively uncopied due to the small number of polycyclic species where the majority of aromatics are benzene derivatives. A representative chromatogram with selected analytes marked is presented in Fig. 3.

A 3 °C min$^{-1}$ temperature gradient was found to be the optimal compromise between peak width, resolution and run time. At this rate, the typical peak width was 10 s in the first dimension. Thus, using a modulation period of 3 s allowed us to obtain the recommended minimum of three modulations per peak (*Murphy, Schure & Foley, 1998*). Due to instrumental complexity and long run times, retention time shifts are commonly observed both in inter-sample and inter-batch runs. Deuterated PAH standards were used to monitor retention times over time. The $^1$D/$^2$D retention time variability is presented in Table 1. Both $^1$D and $^2$D were shown to be stable over the whole analyses run, covering 27 seven injections (nine samples with three extraction replicates) and 7 days.

## Non-target screening and compound identification

Approximately 1,500 compounds were detected in each sample. A data processing workflow was implemented to sort, organize and score the data. A schematic

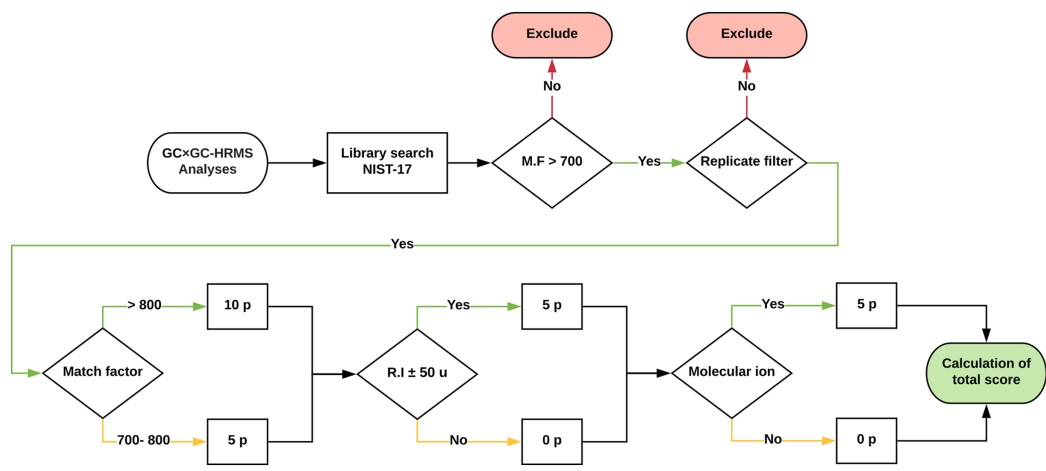

**Figure 4 Schematic of the data processing workflow.** Graphic description of how the data processing workflow scores tentatively identified features for added confidence. The scoring was implemented in a Python script and is available as Supplemental Data.

representation is presented in Fig. 4. The associated files are available as Supplemental Information. The detected features were matched against the NIST EI Mass Spectral Library (2017; NIST v17, Software Version: 2.3). All compounds with a match factor below 700 were removed. To increase the annotation confidence, two additional factors were included; retention index (Kovats) and the presence of the molecular ion. To quantify the identification confidence the following scoring rules were implemented:

1. A match factor above 800 gives a score of 10 points.
2. A match factor between 700 and 800 gives a score of 5 points.
3. A retention index match within 50 units gives a score of 5 points.
4. The detection of the molecular ion within 20 ppm mass accuracy gives a score of 5 points.

A total score of above 10 was (i.e. match factor of a minimum of 700 and either a retention index and or molecular ion match) was classified as a probable match, whereas a total score below 10 was classified as a tentatively identified structure (*Schymanski et al., 2014*). To reduce experimental errors, only features that were present in all three replicates were included in the final table of compounds. Furthermore, all duplicate compounds were removed. This is a crude step that inherently removes, for example, isomeric species which would not be differentiated using automatic library search. However, for our purpose the identification of isomers is not an inherent goal. The aim of the study was to identify the presence of broad compound-class types and dominant species. The correct identification of isomeric species or species with highly similar fragmentation patterns requires manual intervention and is beyond the scope of this study.

The workflow reduced the number of features from approximately **(1)** 1,500 detected to **(2)** 200 library hits with match factor > 700, to **(3)** 50–70 when with duplicates based on name were removed. Ultimately, the merging of inter-sample features resulted in

**Table 2 A list of compounds that were detected in a minimum of six of nine samples.**

| Compound | MS-ready name* | Formula | Mol. weight | XlogP | SMILES | Classification | #Samples |
|---|---|---|---|---|---|---|---|
| (±)-2-Phenylpropanoic Acid, trimethylsilyl ester | 2-phenylpropanoic acid | C9H10O2 | 150.17 | 1.9 | CC(C1=CC=CC=C1)C(=O)O | Aromatic acid | 9 |
| 1-Naphthoic acid, TMS derivative | naphthalene-1-carboxylic acid | C11H8O2 | 172.18 | 3.1 | C1=CC=C2C(=C1)C=CC=C2C(=O)O | Aromatic acid | 9 |
| 3,4-Dimethylbenzoic acid, TMS derivative | 3,4-dimethylbenzoic acid | C9H10O2 | 150.17 | 2.7 | CC1=C(C=C(C=C1)C(=O)O)C | Aromatic acid | 9 |
| Trimethylsilyl 2,3-dimethylbenzoate | 2,3-dimethylbenzoic acid | C9H10O2 | 150.17 | 2.8 | CC1=C(C(=CC=C1)C(=O)O)C | Aromatic acid | 9 |
| Trimethylsilyl 4-propylbenzoate | 4-propylbenzoic acid | C10H12O2 | 164.2 | 3.4 | CCCC1=CC=C(C=C1)C(=O)O | Aromatic acid | 9 |
| 4-tert-Butylphenol, TMS derivative | 4-tert-butylphenol | C10H14O | 150.22 | 3.3 | CC(C)(C)C1=CC=C(C=C1)O | Aromatic alcohol | 9 |
| 3-Methyl-1-cyclohexanecarboxylic acid, trimethylsilyl ester (stereoisomer 2) | 3-methylcyclohexane-1-carboxylic acid | C8H14O2 | 142.2 | 2.1 | CC1CCCC(C1)C(=O)O | Saturated acid | 9 |
| 3-Methylbutanoic acid, TMS derivative | 3-methylbutanoic acid | C5H10O2 | 102.13 | 1.2 | CC(C)CC(=O)O | Saturated acid | 9 |
| Cyclohexaneacetic acid, TMS derivative | 2-cyclohexylacetic acid | C8H14O2 | 142.2 | 2.5 | C1CCC(CC1)CC(=O)O | Saturated acid | 9 |
| Cyclohexanecarboxylic acid, TMS derivative | cyclohexanecarboxylic acid | C7H12O2 | 128.17 | 1.9 | C1CCC(CC1)C(=O)O | Saturated acid | 9 |
| Cyclopentylcarboxylic acid, TMS derivative | cyclopentanecarboxylic acid | C6H10O2 | 114.14 | 1.3 | C1CCC(C1)C(=O)O | Saturated acid | 9 |
| Heptanoic acid, TMS derivative | heptanoic acid | C7H14O2 | 130.18 | 2.5 | CCCCCCC(=O)O | Saturated acid | 9 |
| Nonanoic acid, TMS derivative | nonanoic acid | C9H18O2 | 158.24 | 3.5 | CCCCCCCCC(=O)O | Saturated acid | 9 |
| Octanoic acid, TMS derivative | octanoic acid | C8H16O2 | 144.21 | 3 | CCCCCCCC(=O)O | Saturated acid | 9 |
| 1-Hexadecanol, TMS derivative | hexadecan-1-ol | C16H34O | 242.44 | 7.3 | CCCCCCCCCCCCCCCCO | Saturated alcohol | 9 |
| Benzenepropanoic acid, TMS derivative | 3-phenylpropanoic acid | C9H10O2 | 150.17 | 1.8 | C1=CC=C(C=C1)CCC(=O)O | Aromatic acid | 8 |
| m-Toluic acid, TMS derivative | 3-methylbenzoic acid | C8H8O2 | 136.15 | 2.4 | CC1=CC(=CC=C1)C(=O)O | Aromatic acid | 8 |
| 2,4-Di-tert-butylphenol | 2,4-ditert-butylphenol | C14H22O | 206.32 | 4.9 | CC(C)(C)C1=CC(=C(C=C1)O)C(C)(C)C | Aromatic alcohol | 8 |
| 3-Ethylphenol, TMS derivative | 3-ethylphenol | C8H10O | 122.16 | 2.4 | CCC1=CC(=CC=C1)O | Aromatic alcohol | 8 |
| 4-Isopropylphenol, TMS derivative | 4-propan-2-ylphenol | C9H12O | 136.19 | 2.9 | CC(C)C1=CC=C(C=C1)O | Aromatic alcohol | 8 |
| o-Cresol, TMS derivative | 2-methylphenol | C7H8O | 108.14 | 2 | CC1=CC=CC=C1O | Aromatic alcohol | 8 |
| 2-Methylbutanoic acid, TMS derivative | 2-methylbutanoic acid | C5H10O2 | 102.13 | 1.2 | CCC(C)C(=O)O | Saturated acid | 8 |
| 3-Methylvaleric acid, TMS | 3-methylpentanoic acid | C6H12O2 | 116.16 | 1.6 | CCC(C)CC(=O)O | Saturated acid | 8 |
| 2-Octanol, TMS derivative | octan-2-ol | C8H18O | 130.23 | 2.9 | CCCCCCC(C)O | Saturated alcohol | 8 |

**Table 2** (*continued*)

| Compound | MS-ready name* | Formula | Mol. weight | XlogP | SMILES | Classification | #Samples |
|---|---|---|---|---|---|---|---|
| Benzeneacetic acid, TMS derivative | 2-phenylacetic acid | C8H8O2 | 136.15 | 1.4 | C1=CC=C(C=C1)CC(=O)O | Aromatic acid | 7 |
| Benzenebutanoic acid, TMS derivative | 4-phenylbutanoic acid | C10H12O2 | 164.2 | 2.4 | C1=CC=C(C=C1)CCCC(=O)O | Aromatic acid | 7 |
| m-Cresol, TMS derivative | 3-methylphenol | C7H8O | 108.14 | 2 | CC1=CC(=CC=C1)O | Aromatic alcohol | 7 |
| 2-Hydroxy-4-methylquinoline, trimethylsilyl ether | 4-methyl-1H-quinolin-2-one | C10H9NO | 159.18 | 1.2 | CC1=CC(=O)NC2=CC=CC=C12 | Aromatic amine/alcohol | 7 |
| 4-Methylvaleric acid, TMS derivative | 4-methylpentanoic acid | C6H12O2 | 116.16 | 1.4 | CC(C)CCC(=O)O | Saturated acid | 7 |
| 2-Ethylphenol, TMS derivative | 2-ethylphenol | C8H10O | 122.16 | 2.5 | CCC1=CC=CC=C1O | Aromatic alcohol | 6 |
| 4-Trimethylsilylphenol | phenol | C6H6O | 94.11 | 1.5 | C1=CC=C(C=C1)O | Aromatic alcohol | 6 |
| Benzoic acid, 4-ethoxy-, ethyl ester | ethyl 4-ethoxybenzoate | C11H14O3 | 194.23 | 3.2 | CCOC1=CC=C(C=C1)C(=O)OCC | Aromatic ester | 6 |
| Naphthalene, 1,7-dimethyl- | 1,7-dimethylnaphthalene | C12H12 | 156.22 | 4.4 | CC1=CC2=C(C=CC=C2C=C1)C | Aromatic hydrocarbon | 6 |
| 3-Methyl-1-cyclohexanecarboxylic acid, trimethylsilyl ester (stereoisomer 1) | 3-methylcyclohexane-1-carboxylic acid | C8H14O2 | 142.2 | 2.1 | CC1CCCC(C1)C(=O)O | Saturated acid | 6 |
| Pentanoic acid, TMS derivative | pentanoic acid | C5H10O2 | 102.13 | 1.4 | CCCCC(=O)O | Saturated acid | 6 |
| 2-(1-Adamantyl)ethanol, TMS derivative | 2-(1-adamantyl)ethanol | C12H20O | 180.29 | 3.4 | C1C2CC3CC1CC(C2)(C3)CCO | Saturated alcohol | 6 |

**Notes:**
* The MS-ready name corresponds to the non-derivatized parent compound. Calculated XlogP values were obtained from the PubChem database.
# Samples correspond to the number of samples in which the compound was detected.

120 unique hits across all samples. Of those, 42 had the maximum score of 20 (i.e. match factor > 800, molecular ion detected and retention index within 50 units). Near all (87%) identified compounds are oxygen-containing, with amines, sulfides and hydrocarbons being the remaining constituents. Only 15 compounds (after removal of internal standards and background species) were present in all samples (Table 2).

## DISCUSSION

A few papers have previously described non-target screening of produced water, primarily from the Norwegian continental shelf. Sørensen et al. described the characterization of unpurified DCM extracts with comparisons to the polar and non-polar fractions as isolated by SPE (*Sørensen et al., 2019*). In their study, a high concentration of hydrocarbons including naphthalenes and linear alkanes was detected in the produced water. This is in contrast to our study, where only a few hydrocarbons, in relative trace amounts, were observed as measured by extracted ion chromatograms of known species. The aqueous solubility of saturated hydrocarbons is low. However, BTEX-type (benzene, toluene and xylene) compounds have relatively high solubility but were still not identified.

**Figure 5 The molecular structures of six representative compounds.** The structures were obtained from the subset of compounds detected in a minimum of six out of nine samples. LogP/XLogP values were obtained from the PubChem database. The reported retention times are of the corresponding trimethylsilyl derivatives.

Furthermore, hydrocarbons could be present in the water phase as dispersed oil droplets. We observed that the presence of such droplets in a water sample is heavily dependent on the oil-water ratio and that optical inspection using a microscope was required for their detection. Oil droplets were not observed in the samples included in our study which we believe explain why only small amounts of hydrocarbons were detected. A second explaination could be losses during sample transport and storage, either via volatilization/ diffusion through the plastic container or microbial degradation. However, this cannot be validated without further sampling with more control of the process.

Only a small fraction of the detected compounds were identified with an acceptable confidence level. A total of 36 unique compounds (across all samples) received the maximum identification score, i.e. a match factor >800, retention index match and detection of the molecular ion. The match factor had the most severe impact on feature reduction. Lowering the match factor limit to >600 increased the number of tentatively identified features by approximately 40% (compared to match factor >700). A manual evaluation showed that although several hits were chemically reasonable based on structure and retention index, the lower limit also led to multiple apparent false positives. Added confidence to questionable identities could be obtained by a corroborative study using soft ionization techniques, i.e. chemical ionization or low voltage electron ionization, where the molecular ion is better observed for non-aromatic species.

Looking at the obtained data, two conclusions were made; (1) samples were dominated by oxygenated organics, and (2) sample-to-sample variation was large, both in terms of composition and relative abundance. Oxygenated hydrocarbons form during diagenesis but may also be the result of microbial and or chemical processes during oil production (*Aitken, Jones & Larter, 2004*; *Head, Gray & Larter, 2014*; *Pannekens et al., 2019*). The oxygenation leads to high partitioning into the aqueous phase during oil-water

separation, and these compounds will likely require attention when developing successful water management technologies. Approximately 50% of the compounds were aromatic, primarily benzene-derivatives with few naphthalenes present. The molecular structure of six representative compounds are presented in Fig. 5. Some of the identified compounds are suspected residual production chemicals, e.g. hexadecanol which does not occur naturally in crude oil. Even when comparing two samples that were sampled from the same well and only three days apart, the difference was substantial. As we lack more detailed information on the sampling step, it is difficult to conclude where these differences stem from. The intra-sample variation can be an effect of sampling and oil-to-water ratio in the test separator. The level of dispersed and or layered oil in samples will likely influence the aggregation and solubility of organics in the aqueous phase. A more controlled sampling campaign is required to identify the source of variability.

## CONCLUSIONS

The composition of produced water is highly complex with several unknowns. Our study aimed to narrow this gap by a broad identification of dissolved organics. The implemented identification workflow excluded approximately 95% of the detected compounds (1,000–1,500 per sample), resulting in 50–80 tentatively identified compounds per sample. This demonstrates both the power and pitfalls of non-target screening; more than 100 compounds were identified with an acceptable level of confidence, and more than 1,000 compounds remain unknown. To our knowledge, this is the most comprehensive list of identified compounds in produced water that has been publicly published. However, being a screening study, quantification was not carried out for any compounds. As this is an important factor for environmental assessment, the obtained compound lists should be used to develop targeted methods to look at absolute concentrations. Furthermore, it would be beneficial to reduce the number of unknowns by using complementary techniques (e.g. HPLC-MS and other soft ionization methods), improved custom libraries and in-silico mass spectral prediction.

When performing environmental impact assessments, both structure and concentration have to be taken into consideration (*Tang et al., 2019*). A large part of the detected compounds are present at trace levels. Their concentration will be further reduced at discharge to sea where a rapid dilution to a large body of water occurs. However, potential cocktail effects where the combined effect of a series of micro-pollutants is harmful but not the single species should be accounted for (*Di Poi et al., 2018*). Here, it would be highly beneficial to link non-target screening studies with toxicological measurements. Ultimately, we hope that our study contributes a small piece of the puzzle, and a stepping stone towards further studies to uncover the full picture.

## ACKNOWLEDGEMENTS

The authors are also grateful for the donation of the samples from Total E&P Denmark. Furthermore, the authors kindly acknowledge the Danish Underground Consortium (Total, E&P Denmark, Noreco & Nordsøfonden) for arranging the donation of water samples and then granting permission to publish this work using company samples. The

authors thank Karen L. Feilberg for general discussions. Finally, the authors are grateful for fruitful and constructive feedback on the manuscript from Simon Ivar Andersen.

### Funding
This work was supported by the Danish Hydrocarbon Research & Technology Centre under the Reservoir Fluids Characterization and Produced Water Management programs. The funders had no role in study design, data collection and analysis, decision to publish, or preparation of the manuscript.

### Grant Disclosures
The following grant information was disclosed by the authors:
Danish Hydrocarbon Research & Technology Centre under the Reservoir Fluids Characterization and Produced Water Management.

### Competing Interests
The authors declare that they have no competing interests.

### Author Contributions
- Sofie N. Bergfors conceived and designed the experiments, performed the experiments, analyzed the data, prepared figures and/or tables, authored or reviewed drafts of the paper, and approved the final draft.
- Khoa Huynh conceived and designed the experiments, performed the experiments, prepared figures and/or tables, authored or reviewed drafts of the paper, and approved the final draft.
- Annette E. Jensen conceived and designed the experiments, performed the experiments, authored or reviewed drafts of the paper, and approved the final draft.
- Jonas Sundberg conceived and designed the experiments, performed the experiments, analyzed the data, performed the computation work, prepared figures and/or tables, authored or reviewed drafts of the paper, and approved the final draft.

### Data Availability
Raw, merged and scored feature tables (including mass spectra for each feature) and associated data is available at Zenodo: Sundberg, Jonas. (2020). Non-target screening of organic compounds in offshore produced water by GC×GC-MS (associated data) (Version 1.0) [Data set]. Zenodo. DOI 10.5281/zenodo.4009044.

### Supplemental Information
Supplemental information for this article can be found online at http://dx.doi.org/10.7717/peerj-achem.11#supplemental-information.

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
