# Peer review of "Non-target screening of organic compounds in offshore produced water by GC×GC-MS"

_PeerJ Analytical Chemistry, doi:10.7717/peerj-achem.11_

## Round 0.1 · original submission · Major Revisions

Firstly, please accept my apologies for the delay in processing your manuscript. It proved very difficult to find people who were willing and able to review it but I did not want to make a decision without at least two reviews.

I have now received 2 reviews of the paper and you will see they differ markedly, with one recommending rejection and one suggesting minor revisions.

I feel that the reviewer who suggested rejection, while making some very good points that need to be addressed, did not make any comments that could not be addressed in a revision. Conversely the reviewer who recommended minor changes actually wrote a very detailed review with a lot of comments that need to be addressed. I am therefore recommending major revisions.

Reviewer 1 ·

Basic reporting

The basic reporting is good and the manuscript is structured well. The authors should double check for typos (although they are only minor) - for example, line 25 "furtthermore" should be corrected to "furthermore."

Experimental design

While this is an interesting manuscript and good concept, I think the experimental design is flawed. From my understanding of the manuscript, you are using the NIST library to identify compounds. However, you only use TMS derivatisation to derivatise your samples for running on the GCxGC-MS. This means you are missing many samples or will be unable to identify them using the NIST library. The prime example is fatty acids. Those greater than C12 in length, in particular. While they can and are routinely TMS derivatised, this is not the gold standard for derivatisation and, therefore, much of this data is missing/lacking from the NIST library. These compounds are rountinely derivatised to make methyl esters and it is the MS data from these that are found in the NIST library. For such as scoping study, it would be better to use LC-MS as derivatisation isn't (always) needed.

Validity of the findings

Unfortunately, not all supplemental data was available to me. There was only jpeg images of 2D scans with no figure legends. Therefore, it was difficult to assess many of the points in the manuscript where it said to look at supplementary data.

Additional comments

This is a very interesting study and does show that there is more compounds than we think in produced water.

Line 69 - reference requires a date
Line 76 - "(LLE)" needs to be placed after the word "extraction"
Line 130 - why was 25 ppm chosen? This is not made clear in the manuscript
Line 130 - this information is not in the supplemental data
Line 132 - it says the synthetic water was extracted 4 times, but it doesn't say how. More detailed is required, even if it is just to say "using the above method."
Line 158 - the script isn't in the supplementary information provided
Line 186 - there is no need to say "1.5 mL of n-hexane" because this is actually a false number, because in the actual protocol you are summarising 1 mL is transferred to be derivatised

Figure legends need to be improved to make it easier for the reader to understand what is happening in the figure and they should be able to stand alone of the text.

Reviewer 2 ·

Basic reporting

See comments grouped below.

Experimental design

See comments grouped below.

Validity of the findings

See comments grouped below.

Additional comments

The author study the important problem of produced water, released into the environment after treatment, using GCxGC-MS tools.

- lines 80-91: the authors just mentioned GCxGC, now they write about GC and LC. The authors should clarify if this text (lines 80-91) is about unidimensional or multidimensional GC and LC.
- lines 107-114: were the samples more or less regularly spaced (in time) during this period? Were there some replicate samples?
- line 112: might the plastic from the bottles have affected the results (partitioning of hydrophobic molecules to the plastic?)
- line 116: "extracted in three experimental replicates". Maybe clarify if three "analysis replicates" or "aliquot replicates" are meant. The use of the word "experimental" feels a bit unclear it seems. (Or was there always three replicate samples collected on the platform at each time point?) (If what is described here is the same as at lines 180-181, the text at lines 180-181 is more clear.)
- lines 107-114: it would help to specify at what stage in the process the samples were taken (upon entering the test separator? upon leaving it (effluent)?). What were the different steps of processing before the sample was taken, and are there further steps that are performed on the platform before release into the sea? (It is basically somehow unclear if the sample corresponds to raw water coming out of the well, partly-treated water, or treated water ready for release/disposal). It is also unclear what is meant with "test separator".
additionally: it would help to specify the sample volume.
- line 177 (Figure 1): could the authors specify in the caption to that figure (or in the text) which ones of the bottles on Figure 1 were rejected? Also, could the authors clarify if the nine samples mentioned at line 107 were the number of samples that passed this test, or if (how many?) of these nine samples were excluded because of a visible oil layer.
- line 177 (Figure 1): the caption to Figure 1, it is said that these are the samples "as received". Are these really plastic bottles (line 112)? (I may have a wrong feeling, it looks like glass bottle with a plastic cap...)
- line 179 (Figure 2): the caption to Figure 2 is unclear. What are the two panels?
- lines 183-184: what about small aromatics like benzene, toluene, etc.? Are these not expected to be present? Are these lost (volatilized) during the produced water treatment on the platform?
- lines 195-197: is retention time shifting also a problem (peak deformation, leading to shifting of the peak apex, which is sometime taken as peak position). Or are retention times irrelevant in the authors's analysis? (Line 257 mentions Kovats retention indices, and I think these must be calculated from retention times (?)).
- lines 204-205: what is "synthetic produced water"? clarify the water used..
- line 207: "peak volumes": the authors may like to clarify how peak volumes were determined. This may be relatively software-independent for such well-separated peaks, but in some cases peak volumes may be very dependent on the approach use (e.g. software, baseline approach, etc., see Samanipour et al., 2015).
- line 210: I believe that the authors mean "relative standard", not "relative-standard".
- line 211: "the model compounds": do the author mean "peak volumes" of these compounds(?)
- line 247 (and throughout): "experimental replicates": as written above, is "experimental" really the best choice of word?
- line 253: this is shown in Figure 4, not Figure 3. (By the way, for figure 3, up to the authors, but it might help to annotate the chromatogram to show the positions of a few compounds known to the authors--it seems they describe some in the text. I am personally more familiar with a different set of columns and with crude oil chromatograms, and I'd love to be provided such information on that figure to guide my understanding... (this is a mere suggestion, maybe resulting from my ignorance...))
- around line 294: it would help to provide additional explanation of the two contrasting studies. Was the study by Sorensen et al. for produced water containing droplets? Was the sample at a different stage of processing of the produced water? Was there any difference in the analysis method that can explain that?
- line 318: relative concentrations: it might be appropriate for the authors to provide a quantitative statement? ("high" is very unspecific...)
- line 328: "from the same well and only three days apart,": there is a lack of detail about the sampling times in the methods, which limit the ability to appreciate this statement...
- line 129: "sampling and test separator status" is unclear.
- supplementary files: the chromatograms are given file names that probably pertain to sample ID, but this info is (it seems) not available to the author. Maybe add a table providing understanding for the readers.

---

## Round 0.2 · accepted · Accept

Thank you for making the required changes.

Reviewer 1 ·

Basic reporting

No comment

Experimental design

I think that the reasoning for the experimental design has been addressed adequately.

Validity of the findings

No comment

Additional comments

A very interesting paper. Thank you for clarifying the experimental design and making the suggested changes.